# Intellectual disability-associated gain-of-function mutations in *CERT1* that encodes the ceramide transport protein CERT

**Hiroaki Murakami**[ID][1☯], **Norito Tamura**[2☯], **Yumi Enomoto**[3], **Kentaro Shimasaki**[2], **Kenji Kurosawa**[ID][1]*, **Kentaro Hanada**[ID][2]*

**1** Division of Medical Genetics, Kanagawa Children's Medical Center, Yokohama, Kanagawa, Japan,
**2** Department of Biochemistry & Cell Biology, National Institute of Infectious Diseases, Tokyo, Japan,
**3** Clinical Research Institute, Kanagawa Children's Medical Center, Yokohama, Kanagawa, Japan

☯ These authors contributed equally to this work.
* kkurosawa@kcmc.jp (KK); hanak@nih.go.jp (KH)

**Data Availability Statement:** As described under "Data Availability" of the manuscript, the original uncropped and unadjusted images for all blot are

## Abstract

Intellectual disability (ID) is a developmental disorder that includes both intellectual and adaptive functioning deficits in conceptual, social, and practical domains. Although evidence-based interventions for patients have long been desired, their progress has been hindered due to various determinants. One of these determinants is the complexity of the origins of ID. The ceramide transport protein (CERT) encoded by *CERT1* mediates inter-organelle trafficking of ceramide for the synthesis of intracellular sphingomyelin. Utilizing whole exome sequencing analysis, we identified a novel CERT variant, which substitutes a serine at position 135 (S135) for a proline in a patient with severe ID. Biochemical analysis showed that S135 is essential for hyperphosphorylation of a serine-repeat motif of CERT, which is required for down-regulation of CERT activity. Amino acid replacements of S135 abnormally activated CERT and induced an intracellular punctate distribution pattern of this protein. These results identified specific ID-associated *CERT1* mutations that induced gain-of-function effects on CERT activity. These findings provide a possible molecular basis for not only new diagnostics but also a conceivable pharmaceutical intervention for ID disorders caused by gain-of-function mutations in *CERT1*.

## Introduction

ID is a developmental disorder that includes both intellectual and adaptive functioning deficits in conceptual, social, and practical domains [1]. ID has an overall general population prevalence of ~1%, which varies by age [1]. Although evidence-based interventions for patients have long been desired, their progress has been hindered due to various determinants [2, 3]. One of these determinants is the complexity of the origins of IDs, which include: inborn genomic mutations, malconditions (e.g., hypoxia) in the fetal period, detrimental stresses (e.g., physical and/or mental abuse) in the infant period, an inferior social environment (e.g., unavailability of education) in childhood, and other factors.

presented in S5 Fig. In addition, other unprocessed images and raw data for quantitative results have been posted at a public data repository (https://doi.org/10.6084/m9.figshare.12830852).

**Funding:** N.T.: Grant No. 19J01972, Japan Society for the Promotion of Science (https://www.jsps.go.jp/english/index.html). K.K.: Grant No. 16H06279, Japan Society for the Promotion of Science (https://www.jsps.go.jp/english/index.html). K.H.: Grant No. JP17H06417, Ministry of Education, Culture, Sports, Science and Technology (https://www.mext.go.jp/en/index.htm). K.H.: Grant No. JP20gm0910005j0006, Japan Agency for Medical Research and Development (https://www.amed.go.jp/en/index.html). The funders had no role in study design, data collection and analysis, decision to publish, or preparation of the manuscript.

**Competing interests:** The authors have declared that no competing interests exist.

Recent advances in human medical genomics have facilitated the identification of causative mutations of various genetic diseases, which may provide rationales for personally tailored interventions for patients with specific mutations. Whole exome sequencing (WES) analysis has been applied to ID studies, and has identified various *de novo* mutations that are strongly associated with the induction of IDs [4–9]. Recently, several independent studies found that *de novo* mutations in *CERT1* can induce IDs [4–7]. The lipid ceramide is synthesized in the endoplasmic reticulum (ER) and the ceramide transport protein CERT encoded by *CERT1* transports ceramide from the ER to the *trans*-Golgi regions. Once localized to the *trans*-Golgi regions, ceramide is metabolized to sphingomyelin (SM), which is one of the major phospholipid types in mammalian cells [10, 11]. Previous studies demonstrated that a serine-repeat motif (SRM) of CERT undergoes multiple phosphorylations, which down-regulate CERT activity [12]. Intriguingly, various examples of ID-related *de novo* mutations in *CERT1* were mapped to the region encoding the SRM [4, 5, 7]. These mutations are expected to inhibit SRM hyperphosphorylation, thereby disrupting the post-translational down-regulation of CERT. However, it has not been shown whether these mutations affect the phosphorylation of the CERT SRM and render the mutated CERT abnormally active.

In this study, we identified a novel *de novo* missense mutation, which substitutes a serine at position 135 for a proline (S135P), in *CERT1* (NM_0031361.3:c.403T>C:p.[Ser135Pro]) by trio WES analysis of a patient with a severe ID and systemic symptoms. Moreover, we showed that this novel mutation and other previously reported mutations inhibited the hyperphosphorylation of the CERT SRM, which resulted in abnormally activated CERT.

## Materials and methods

### Study subjects and ethics statement

This study was approved by the Review Board and Ethics Committee of the Kanagawa Children's Medical Center (Approval number: 118–17). Written informed consent (as outlined in PLOS consent form) was obtained from the patient's parents, which included consent for the pictures appearing in the manuscript. The biochemical and cell biological analyses of LCLs were also approved by the Medical Research Ethics Committee of the National Institute of Infectious Diseases (Approval number: 1030).

### Mutation screening and assessment

Genomic DNA was purified from the peripheral blood of the patient and her parents utilizing a QIAcube (QIAGEN, Hilden, Germany) according to the manufacturer's instructions. Initially, we performed a single WES analysis of the patient. Purified DNA was enriched using a SureSelectXT Human All Exon Enrichment kit (Agilent Technologies, Santa Clara, CA, USA), and sequenced on a HiSeq platform (Illumina Inc., San Diego, CA, USA). Sequence data alignment, variant calling, and variant annotation were performed as described previously [13]. Additionally, we conducted WES analysis of the parent's genomic DNA, which was sequenced using a NovaSeq platform (Illumina Inc., San Diego, CA, USA). In order to evaluate the variants identified in the patient, we used the Exome Aggregation Consortium (ExAC) (http://exac.broadinstitute.org.), 1000 Genomes Project (https://www.internationalgenome.org/), and jMorp (https://jmorp.megabank.tohoku.ac.jp/202001/variants) as the control healthy populations. All the detected variants were classified according to the 2015 ACMG guidelines [14]. The prediction of pathogenicity was performed using the Combined Annotation Dependent Depletion (CADD) (https://cadd.gs.washington.edu/), Sorting Intolerant From Tolerant (SIFT) (http://sift.jcvi.org/), and Polymorphism Phenotyping v2 (PolyPhen-2) tools (http://genetics.bwh.harvard.edu/pph2/). The current Human Genome Variation Society standards

were employed for the mutation nomenclature, which was confirmed with Mutalyzer software (http://mutalyzer.nl/).

## Establishment of immortalized B-cell lines

Immortalized B-cell lines were established by Epstein-Barr virus transformation of fresh peripheral blood lymphocytes of the patient and her parents as described previously [15].

## Cell culture

Immortalized B-cells were cultured in Roswell Park Memorial Institute (RPMI) 1640 medium (Gibco, Grand Island, NY, USA) supplemented with 10% fetal bovine serum (FBS; Sigma-Aldrich, St. Louis, MO, USA) and were grown in a 5% $CO_2$ incubator. Pseudodiploid HCT116 cells were obtained from the RIKEN cell bank [16], These cells were cultured in low glucose Dulbecco's Modified Eagle Medium (DMEM; WAKO, Osaka, Japan) supplemented with 10% FBS and were grown in a 5% $CO_2$ incubator. For lysenin treatment, cells were incubated in serum-free DMEM containing 250 ng/ml of lysenin for 1 h.

## Retroviral preparation and establishment of stable cell lines

Stable cell lines were generated using a previously described method, with modifications [17]. Briefly, Plat-E cells were transiently transfected with pLP-VSV-G (Thermo Fisher Scientific, San Jose, CA, USA) and retroviral vectors using the FuGENE6 transfection reagent (Promega, Madison, WI, USA). After 48 h, the culture medium containing retrovirus particles was collected and filtered through a 0.45 μm filter unit (Merck Millipore, Darmstadt, Germany). Next, the HCT116 cells were incubated with the retrovirus particles and 8 μg/ml of polybrene (Sigma-Aldrich, St. Louis, MO, USA). The uninfected cells were removed by treatment with 2 μg/ml blasticidin S (Kaken Pharmaceutical, Tokyo, Japan).

## Plasmids, antibodies, and reagents

cDNAs encoding the full-length human CERT (NP_112729) and its variants were amplified by polymerase chain reaction (PCR) and subcloned into the pMXs-IB [18] or pCX4 [19] backbone vectors together with the human influenza HA tag or the mVenus tag. The plasmid pSpCas9 (BB)-2A-GFP (pX458) (plasmid # 48138; Addgene, Cambridge, MA, USA) was used for the establishment of the KO cell line. For Western blotting analysis, the following primary antibodies were used: rabbit polyclonal anti-CERT (ab72536; Abcam, Cambridge, MA, USA), anti-HA conjugated with horseradish peroxidase (HRP; clone 3F10; Roche, Mannheim, Germany) as well as mouse monoclonal anti-β-actin (sc-47778; Santa Cruz Biotechnology, Santa Cruz, CA, USA) and the following secondary antibodies were used: anti-mouse (NA934; Cytiva, Marlborough, MA, USA) and anti-rabbit (172–1011; Bio-Rad Laboratories, Hercules, CA, USA) HRP-conjugated immunoglobulin G (IgG). For immunofluorescence, the following primary antibodies were used: mouse monoclonal anti-GM130 (610822; BD Bioscience, San Jose, CA, USA) as well as rabbit polyclonal anti-VAP-A (HPA009174; Sigma-Aldrich, St. Louis, MO, USA) and the following secondary antibodies were used: Alexa Fluor 594-conjugated goat anti-mouse IgG (A11032; Thermo Fisher Scientific, San Jose, CA, USA) and Alexa Fluor 647-conjugated goat anti-rabbit IgG (A32795; Thermo Fisher Scientific, San Jose, CA, USA).

## Lysenin tolerance assay

Lysenin purified from the coelomic fluid of the earthworm *Eisenia foetida* was a gift from Yoshiyuki Sekizawa (Zenyaku Kogyo Co., Tokyo, Japan) [20]. Cells were seeded at an initial density of $1.0 \times 10^4$ cells/well in a 24-well plate and were grown under normal conditions overnight. The cells were then incubated in serum-free DMEM containing 250 ng/ml of lysenin for 1 h. The cell viability was measured with LDH cytotoxicity assay kit (Nacalai Tesque, Kyoto, Japan) according to the manufacturer's protocol. In our assay, the cell viability ("survival ratio after lysenin-treatment") was defined as the following formula: 1-(A1-A2)/(A3-A2). Here, A1, A2, and A3 represent LDH activity (measured by absorbance) from lysenin-treated cells, untreated control cells, and detergent-treated cells, respectively. The LDH activity from culture medium that was not used to grow cells was measured and subtracted as a background control. The luminescence was measured with an iMark microplate reader (Bio-Rad Laboratories, Hercules, CA, USA).

## Immunofluorescence

Cells grown on coverslips (Matsunami Glass, Osaka, Japan) were washed with PBS and fixed with 4% paraformaldehyde (Mildform 10N; WAKO, Osaka, Japan) for 15 min at room temperature. The fixed cells were then permeabilized with 0.1% TritonX-100 in PBS for 5 min, blocked with 3% bovine serum albumin (Sigma-Aldrich, St. Louis, MO, USA) in PBS, and incubated with specific primary antibodies for 1 h. After washing with PBS, the cells were incubated with either Alexa Fluor 594-conjugated goat anti-mouse IgG or Alexa Fluor 647-conjugated goat anti-rabbit IgG secondary antibodies for 1 h. The cells were viewed using a confocal laser microscope (LSM700; Carl Zeiss, Jena, TH, Germany) equipped with a $100 \times$ oil immersion objective lens (1.46 NA; Carl Zeiss, Jena, TH, Germany), and the view was captured with the ZEN software (Carl Zeiss, Jena, TH, Germany). Photoshop CS6 software (Adobe Systems Corporation, San Jose, CA, USA) was used for the final presentation of the images.

## λPPase treatment

Cells were lysed with the lysis buffer lacking phosphatase inhibitor cocktails, and the protein concentration of each lysate was determined with the BCA method. Next, the lysates (50 μl) were incubated with 400 units of λPPase (New England Biolabs, Beverly, MA, USA) at 30˚C for 15 min.

## Establishment of *CERT1* KO cell lines

The CRISPR guide RNA (gRNA) sequence designed to target the human *CERT1* gene was cloned into the pX458 vector. The target sequence is 5'-GCTCTGATTATCCGACATGG-3'. HCT116 cells were transfected with the pX458 vector harboring the above gRNA with the ViaFect Transfection Reagent (Promega, Madison, WI, USA). After 48 h, green fluorescent protein (GFP)-positive cells were isolated using a cell sorter (BD FACSMelody; BD Bioscience, San Jose, CA, USA) and single clones were obtained. The cell clones containing the desired mutations in both *CERT1* alleles were identified by genomic DNA sequencing (S3A Fig), utilizing the genomic PCR primers 5'-AAATTGGCATCGAGGGGGCTAAGTTCGGG-3', and 5'-CTCATCCCTAGTCGCTGCAGCAACAC-3'.

## Western blotting

Cells were lysed with a lysis buffer [50 mM Tris-HCl (pH 7.5), 150 mM NaCl, 1 mM EDTA, 1% Triton X-100, complete EDTA-free protease inhibitor cocktail (Roche, Mannheim,

Germany), and phosphatase inhibitor cocktails 2 and 3 (Sigma-Aldrich, St. Louis, MO, USA)] and centrifuged at $20,000 \times g$ for 20 min at 4°C. The supernatants were collected, and the protein concentrations were quantitated with the bicinchoninic acid (BCA) method (Thermo Fisher Scientific, San Jose, CA, USA) using bovine serum albumin as the standard. The lysates were then solubilized with immunoblot sample buffer [46.7 mM Tris-HCl (pH 6.8), 5% glycerol, 1.67% SDS, 1.55% dithiothreitol, and 0.003% bromophenol blue]. Next, the samples were separated by SDS-PAGE, transferred to an Immobilon-P polyvinylidene difluoride membrane (Millipore, Darmstadt, Germany), and blotted with primary and secondary antibodies. Each protein signal was detected with the Immobilon Western Chemiluminescent HRP Substrate (Millipore, Darmstadt, Germany). The signal intensities were captured with the LuminoGraph image analyzer (ATTO, Tokyo, Japan) and the images were processed using Photoshop CS6 software (Adobe Systems Corporation, San Jose, CA, USA).

## Sphingolipid labeling with radioactive serine and thin layer chromatography (TLC)

The metabolic labeling of sphingolipids with radioactive serine was performed as described previously [21]. Briefly, LCLs or HCT116 cells ($5.0 \times 10^5$–$1.0 \times 10^6$ cells) were cultured in a 12- or 6-well plate, respectively. After overnight incubation under normal conditions, LCLs or HCT116 cells were incubated with 1% Neutridoma-SP (Roche, Mannheim, Germany) and 18.5 kBq of L-[U-$^{14}$C]serine (Moravek Biochemicals, Brea, CA, USA) for 16 h in RPMI medium supplemented with 10% FBS or in serum-free low glucose DMEM, respectively. After the labelled cells were washed with phosphate buffered saline (PBS), they were lysed with a 0.1% sodium dodecyl sulfate (SDS) solution. The lipids were extracted from the lysate and separated by TLC with the solvent methyl acetate/n-propanol/chloroform/methanol/0.25% KCl (50/50/50/20/18, vol/vol) as described previously [22]. The radioactive lipids on TLC plates (Millipore, Darmstadt, Germany) were detected using an image analyzer (Typhoon FLA 7000; GE Healthcare, Madison, WI, USA). The intensities of the radioactive lipid bands visualized on the TLC plates were determined by densitometric scanning using ImageJ software (National Institutes of Health). The intensity levels of SM were normalized to phosphatidylserine in order to compare the relative levels of labelled SM among different cell lines.

## Statistics

For data analysis of biochemical assays, a one-way analysis of variance (ANOVA) followed by Tukey's or Dunnett's test was conducted. Descriptive statistics are presented as the mean ± standard error of the mean (SEM) from technical replicates. Results were considered significant at the 95% significance level ($p < 0.05$). The statistical analysis was performed using R software (R Core Team).

## Results

### Phenotypic details of a proband

The proband was a 23-year-old girl and the second child of healthy nonconsanguineous parents. She was born after a full-term delivery with normal birth parameters after an uneventful pregnancy (Fig 1A). Her developmental milestones were severely delayed. She attained head control at the age of 6 months. She could not roll over but could sit with support at the age of 4 years. At 5 years of age, she developed epilepsy triggered by an influenza virus infection and began taking an anti-epilepsy drug. Brain magnetic resonance imaging (MRI) at 5 years of age showed delayed myelination of the cerebrum and hypoplasia of the corpus

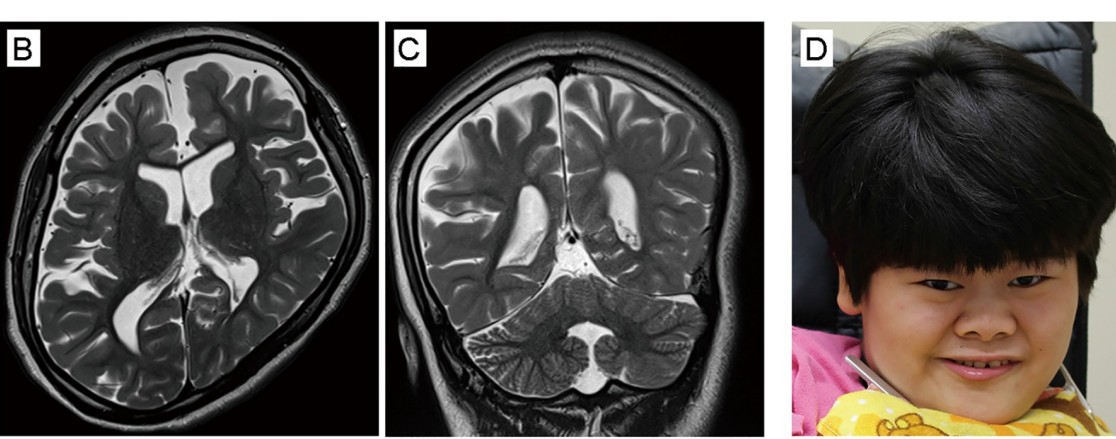

| A | | | |
|---|---|---|---|
| Gestational age | 37 weeks and 5 days | | |
| Birth parameters | Weight 2466 g (-0.9 SD) | Height 44.0 cm (-1.8 SD) | OFC 33.2 cm (+0.2 SD) |
| Systemic complications | Growth<br>Central nerve system<br>Muscular tonus<br>Skeletal system | Failure to thrive<br>Brain abnormalities<br>Hypotonia<br>Scoliosis and torticolis | |

**Fig 1. Clinical features of a patient with severe ID.** (A) Perinatal information and systemic symptoms of the patient. SD, standard deviation. (B) Brain MRI (T2-weighted imaging) of the patient at 23 years of age (axial imaging). The MRI revealed general cerebral atrophy, especially in the frontal lobe, and hypoplasia of the corpus callosum. (C) Coronal MRI imaging. (D) Patient photograph showing the coarse face, thick eyebrows, hypotelorism, long palpebral fissures, midface hypoplasia, bulbous nose, wide ala nasi, short philtrum, thin upper lip, and hypertrichosis.

callosum. Her intelligence quotient was less than 35 at 6 years of age and she could not stand alone nor speak any meaningful words. The progression of her leukodystrophy, especially in the frontal lobe, was confirmed by MRI at 23 years of age (Fig 1B and 1C). On physical examination, she exhibited several systemic complications and distinctive facial features (Fig 1A and 1D) (for additional clinical information, see S1 Table).

## Mutation screening and assessment

At 20 years of age, we conducted trio WES analysis of the patient and her parents and identified a novel heterozygous missense variant in *CERT1* (NM_001130105.1:c.787T>C:p.[Ser263-Pro]) (Fig 2A and 2B). Trio Sanger sequencing confirmed that this was a *de novo* variant (Fig 2C) that was absent in the control healthy population found in the genome aggregation database (gnomAD; https://gnomad.broadinstitute.org/) and the Japanese multi omics reference panel (jMorp; https://jmorp.megabank.tohoku.ac.jp/202001/variants). *In silico* pathogenicity prediction tools indicated strong pathogenicity of this variant (Table 1). According to the American College of Medical Genetics and Genomics (ACMG) guidelines [14], this variant was pathogenic. The *CERT1* probability score that predicts the presence of autosomal dominant changes was high (pAD = 0.849) [23], while the *CERT1* probability score for the loss-of-function intolerant was not high (pLI = 0.67, observed/expected metric = 0.21) (https://gnomad.broadinstitute.org/gene/ENSG00000113163?dataset=gnomad_r2_1). These data indicated that *CERT1* likely acquires pathogenicity by dominant active or negative mechanisms and is likely not a haploinsufficiency gene. This is consistent with the fact that all the previously

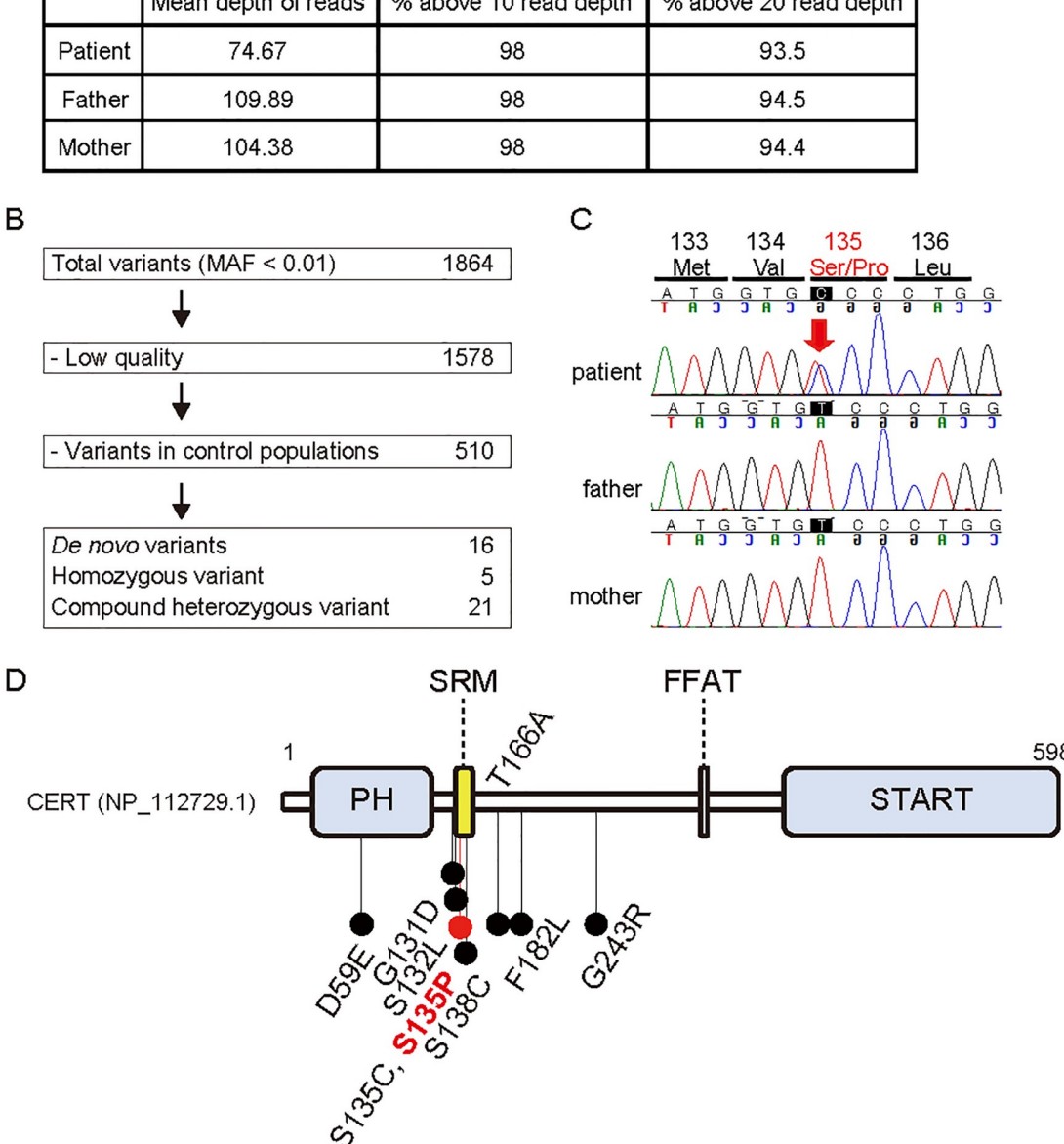

**Fig 2. An ID-associated *de novo* *CERT1* mutation.** (A) WES sequencing quality. (B) Filtering pipeline of variants identified by trio WES. A total of 1864 variants with a minor allele frequency (MAF) of less than 0.01 were found in the patient. After excluding low-quality variants and variants found in the control healthy population, 16, 5, and 21 variants were identified as *de novo* heterozygous, homozygous, and compound heterozygous, respectively. (C) Trio Sanger sequence of the *CERT1* variant (NM_0031361.3:c.403T>C:p.[Ser135Pro], Chr5[GRCh37]:g.74722249A>G) identified in our patient. (D) Variant distribution of the CERT protein structure (NP_112729.1), which includes the PH domain, SRM, FFAT motif, and steroidogenic acute regulatory protein-related lipid transfer (START) domain.

reported *CERT1* variants that cause ID or autism spectrum disorder (ASD) were missense variants, and not truncation variants (Fig 2D and Table 1) [4–7, 9, 24, 25].

## Predominant form of *CERT1* transcripts

*CERT1* contains 20 exons and produces at least three alternatively spliced variants, which we have named isoforms 1–3 (S1 Fig). The isoform 3 transcript is the longest *CERT1* mRNA

**Table 1. Pathogenicity assessment of *CERT1* variants identified in the present and previous studies.**

| References | Variants | Isoform 3 => 1 | Inheritance | Frequency in control cohort | CADD | SIFT | PolyPhen2 | Criteria in ACMG 2015 |
|---|---|---|---|---|---|---|---|---|
| Takata et al. 2018 | c.561T>G:p.(Asp187Glu) | p.(Asp59Glu) | *de novo* | Nothing | 12.31 | Tolerated | 0.03 Benign | Likely pathogenic PS2, PM2 |
| Kosmicki et al. 2017 | c.776G>A:p.(Gly259Asp) | p.(Gly131Asp) | *de novo* | Nothing | 28.5 | Damaging | 1.00 Probably damaging | Likely pathogenic PS2, PM2, PP3 |
| Fitzgerald et al. 2015 | c.779C>T:p.(Ser260Leu) | p.(Ser132Leu) | *de novo* | Nothing | 32.0 | Damaging | 0.998 Probably damaging | Likely pathogenic PS2, PM1, PM2, PP3 |
| **This study** | c.787T>C:p.(Ser263Pro) | p.(Ser135Pro) | *de novo* | Nothing | 28.0 | Damaging | 0.999 Probably damaging | Pathogenic PS2, PM1, PM2, PM5, PP3 |
| Lelieveld et al. 2017 | c.788C>G:p.(Ser263Cys) | p.(Ser135Cys) | *de novo* | Nothing | 28.2 | Damaging | 0.999 Probably damaging | Likely pathogenic PS2, PM1, PM2, PP3 |
| de Ligt et al. 2012 | c.797C>G:p.(Ser266Cys) | p.(Ser138Cys) | *de novo* | Nothing | 28.3 | Damaging | 1.000 Probably damaging | Likely pathogenic PS2, PM1, PM2, PP3 |
| Wang et al. 2016 | c.880A>G:p.(Thr294Ala) | p.(Thr166Ala) | *de novo* | Nothing | 22.8 | Damaging | 0.287 Benign | Likely pathogenic PS2, PM2 |
| Wang et al. 2016 | c.928T>C:p.(Phe310Leu) | p.(Phe182Leu) | *de novo* | Nothing | 23.4 | Damaging | 0.01 Benign | Likely pathogenic PS2, PM2 |
| Hamdan et al. 2014 | c.1111G>A:p.(Gly371Arg) | p.(Gly243Arg) | *de novo* | Nothing | 32.0 | Damaging | 1.000 Probably damaging | Likely pathogenic PS2, PM2, PP3 |

CADD; Combined Annotation Dependent Depletion, SIFT; Sorting intolerance from tolerance, PolyPhen-2; Polymorphism phenotyping version 2; ACMG; American college of medical genetics and genomics; PS, strong pathogenicity; PM, moderate pathogenicity; PP, supporting pathogenic

isoform that encodes a putative very long variant of CERT (CERT/VL). However, RNA sequencing data from human tissues found in the Genotype-Tissue Expression project (GTEx; https://gtexportal.org/home/) indicated that the expression of exon 1 from *CERT1* (that is present in isoform 3) is absent or very rare in all human tissues except for the testis (S2 Fig). In contrast, the *CERT1* isoform 1 transcript consisting of exons 2–11 and 13–19 was the most abundant type. Moreover, the isoform 2 transcript consisting of exons 2–19, which encodes a long variant of CERT (CERT/L) is expressed but at lower levels than isoform 1 in most tissue types (S2 Fig). These expression levels are consistent with a previous mRNA hybridization study [26]. Hereafter, we refer to isoform 1 as the predominant *CERT1*-derived transcript, which is translated into the 598 amino-acid protein CERT (NP_112729.1) (S1 Fig). Our ID patient contained a mutation in isoform 1, which is translated into a variant form of CERT that substituted a serine at position 135 for a proline (S135P) in the SRM.

## Characterization of CERT in trio-derived permanent cell lines

Our previous study demonstrated that phosphorylation of multiple S/T residues of the CERT SRM down-regulated the activity of CERT. Moreover, a S132A substitution in CERT renders the protein constitutively active [12]. Recent human WES studies have revealed various examples of missense mutations (e.g., S132L, S135C and S138C) that generated amino acid replacements in the SRM of CERT and induced inherent ID disorders [4, 5, 7]. However, it has not been determined whether the ID-associated missense mutations in *CERT1* affected the

phosphorylation status of the CERT SRM. Thus, we examined this phenomenon using trio-derived B-cell lines that were established by Epstein-Barr virus infection.

A maximum of 10 phosphorylations occur on the SRM of CERT and the SRM-hyperphosphorylated form of CERT can be separated from its dephosphorylated and/or hypophosphorylated (de/hypophosphorylated) forms in SDS-PAGE [12]. The hyperphosphorylated form of CERT is far more abundant than the de/hypophosphorylated forms in CHO cells [10], HeLa cells [12], and human colon cancer-derived HCT116 cells (S3B Fig). Moreover, Western blotting of lymphoblastoid cell (LCL) extracts derived from our ID patient and her parents showed that the hyperphosphorylated form of CERT was more abundant than the de/hypophosphorylated forms in both the mother and father, while the de/hypophosphorylated forms were more abundant than the hyperphosphorylated form in the patient (Fig 3A). These results are in line with the finding that the patient's mutation is heterozygous and suggest that the CERT S135P mutant is incapable of becoming hyper-phosphorylated. A faint band comprising the hyperphosphorylated form of CERT/L was also observed in all trio LCLs (Fig 3A). The de/hypophosphorylated forms of CERT/L were not clearly observed, presumably due to their size overlap with the hyperphosphorylated form of CERT (Fig 3A). When cell lysates were treated with the lambda protein phosphatase (λPPase), the Western blotting patterns of the trio were converted into two bands. The upper and lower bands correspond to the de-phosphorylated forms of CERT/L and CERT, respectively (Fig 3B). The phosphatase treatment experiment validated the accuracy of the assignments of the various CERT forms in the untreated samples. It should also be noted that no protein derived from the putative *CERT1* isoform 3 transcript was detected in LCLs (Fig 3A and 3B) and HCT116 cells (S3B Fig).

In order to explore whether the ID-associated mutations rendered CERT abnormally active, we conducted metabolic labelling of sphingolipids with radioactive serine. All the types of sphingolipids and several types of glycerophospholipids use serine as their metabolic precursor and CERT activity in living cells can be semi-quantitatively monitored by the level of *de novo* SM synthesis [10]. Under our experimental conditions, no significant difference was observed in the *de novo* SM synthesis among the trio LCLs, although the synthesis of SM in patient-derived LCLs was slightly higher than that in the parents (Fig 3C).

## Characterization of CERT S135 mutants in the absence of the endogenous wild-type CERT

In order to confirm that the CERT S135P mutant protein is incapable of being hyperphosphorylated, we characterized CERT mutants in the absence of the endogenous wild-type (WT) CERT. For this purpose, we disrupted both alleles of *CERT1* in HCT116 cells, and various constructs encoding CERT mutants tagged with influenza hemagglutinin (HA) or monomeric Venus (mVenus) were stably expressed in the HCT116 *CERT1* knockout (KO) cells. Like the endogenous CERT in HCT116 cells, the ectopically expressed HA- or mVenus-tagged WT CERT displayed a major hyperphosphorylated form (Fig 4A and S3C Fig). In contrast, all the ectopically expressed HA- or mVenus-tagged CERT S135P, S135C, and S135A mutant constructs only displayed de/hypophosphorylated forms, which indicated that these mutants were not hyperphosphorylated on the SRM.

Next, we conducted metabolic labeling of lipids with radioactive serine. Metabolic labelling of SM, but not other sphingolipids (i.e., ceramide and glucosylceramide) nor glycerophospholipids (i.e., phosphatidylserine and phosphatidylethanolamine), was markedly reduced in *CERT1* KO cells, compared with parental HCT116 cells (Fig 4B and S3D Fig). This reduction was restored to the parental control levels when either mVenus-tagged WT CERT or CERT/L was expressed in the *CERT1* KO cells (S4 Fig). When the mutant CERT S135A, S135C, or

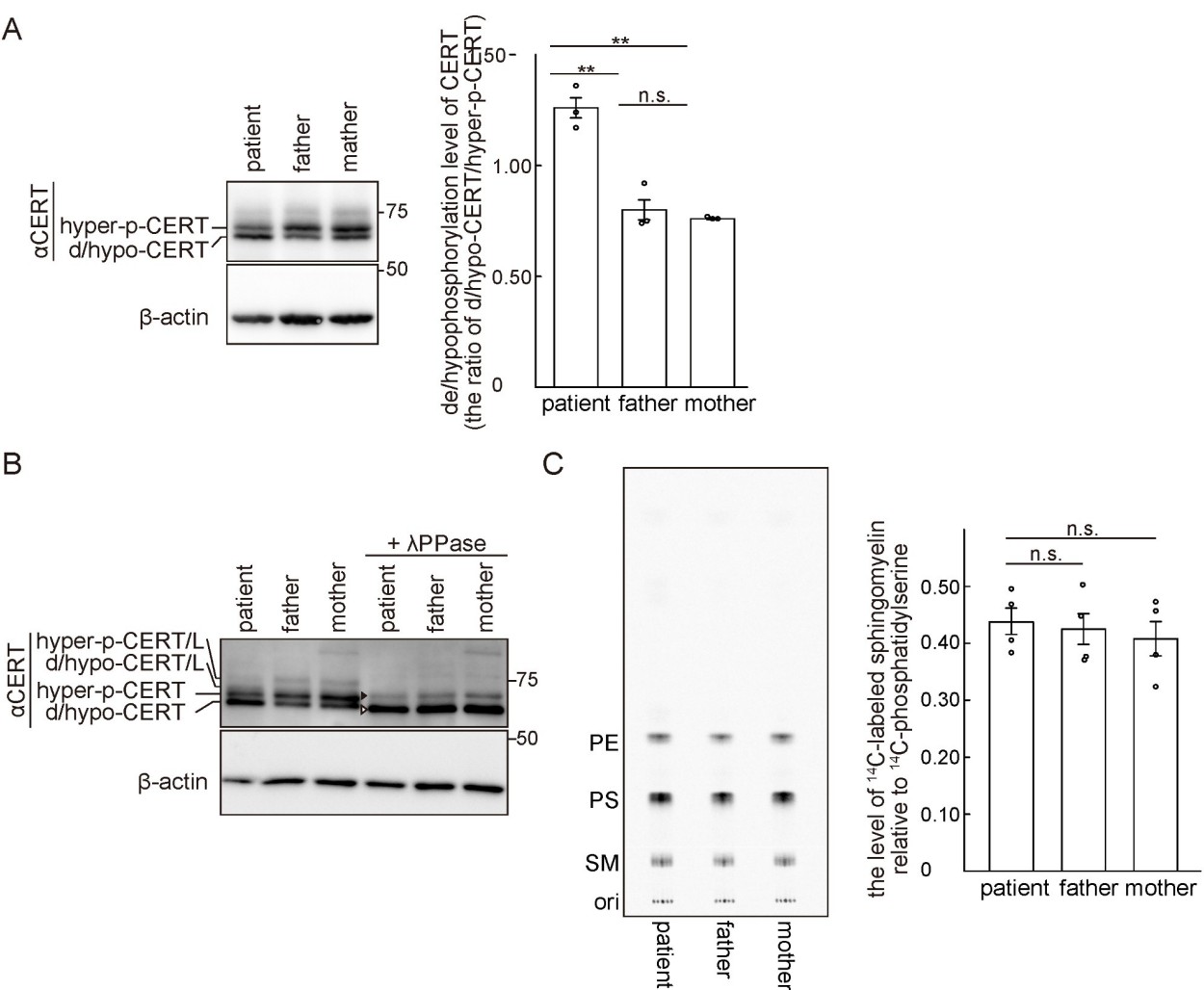

**Fig 3. The ID-associated *de novo* mutation in *CERT1* affects the phosphorylation status of CERT in LCLs.** (A) The trio-derived LCLs were analyzed by Western blotting with the indicated primary antibodies. Hyperphosphorylated (hyper-p-) and de/hypophosphorylated (d/hypo-) CERT are shown (left). The de/hypophosphorylated levels of CERT in the trio-derived LCLs were quantified by densitometric scanning of the band intensities (right). The data comprise the mean ± SEM; n = 3 (**, $p < 0.01$; n.s., not significant). (B) Trio LCL lysates were incubated with or without λPPase and analyzed by Western blotting with the indicated primary antibodies. The white and black arrowheads represent completely de/hypophosphorylated CERT and CERT/L, respectively. (C) Trio LCLs were cultured with L-[U-$^{14}$C]serine for 16 hr. Metabolically labelled lipids separated on a TLC plate were visualized (representative image, left) and labelled SM was quantified (right). PE, phosphatidylethanolamine; PS, phosphatidylserine. The data comprise the mean ± SEM; n = 4.

S135P constructs were expressed, the levels of labelled SM were significantly higher than the level observed in the WT CERT rescued cells, despite lower expression levels of these CERT mutants compared to WT CERT levels (S3C Fig). The sensitivity to lysenin, an SM-binding cytolysin, was monitored as another measure of SM synthesis. Cells expressing the CERT mutants exhibited higher lysenin sensitivity (Fig 4C), which indicated that the intracellular CERT S135A, S135C, and S135P mutants exhibited higher activity than the WT CERT.

When mVenus-tagged CERT variants were expressed in HCT116 *CERT1* KO cells, the WT CERT was diffused throughout the cytoplasm with a partial enrichment in the peri-nuclear regions, which are adjacent to a 130 kDa *cis*-Golgi matrix protein (GM130) (Fig 4D). Interestingly, all the CERT S135 variants exhibited a punctate distribution pattern, and part of the

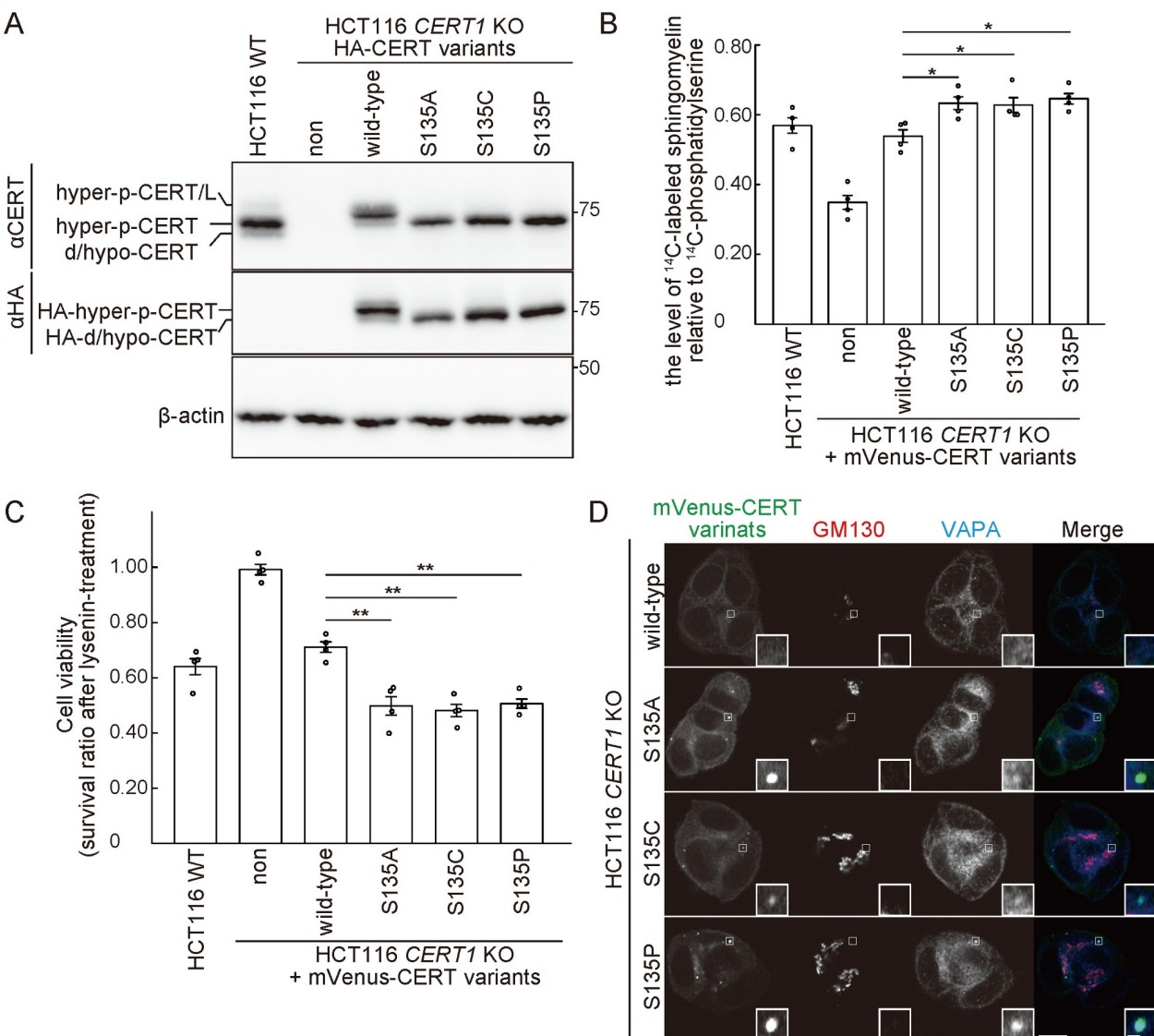

**Fig 4. The CERT S135P mutant enhances *de novo* SM synthesis in HCT116 cells.** (A) WT (HCT116 WT) and *CERT1* KO HCT116 cells stably expressing various HA-CERT constructs were analyzed by Western blotting. (B) WT and *CERT1* KO HCT116 cells stably expressing various mVenus-CERT constructs were cultured with L-[U-$^{14}$C]serine for 16 hr. Metabolically labeled lipids were analyzed by TLC. The data comprise the mean ± SEM; n = 4 (*, $p < 0.05$). (C) WT and *CERT1* KO HCT116 cells stably expressing the indicated mVenus-CERT constructs were cultured with 250 ng/ml of lysenin for 1 hr. The cell viability was measured with a lactate dehydrogenase (LDH) cytotoxicity assay. The data comprise the mean ± SEM; n = 4 (*, $p < 0.05$; **, $p < 0.01$). (D) *CERT1* KO HCT116 cells stably expressing the indicated mVenus-CERT constructs were analyzed by immunofluorescence microscopy. The data are representative of two independent experiments. Areas enclosed by rectangles are enlarged (insets). Scale bars, 10 μm and 1 μm (inset).

puncta co-stained with the vesicle-associated membrane protein-associated protein (VAP)-A (Fig 4D).

## The abnormal phenotypes of CERT S135 mutants require their Golgi- and ER-binding activities

CERT is recruited to the Golgi apparatus via binding of its pleckstrin homology (PH) domain to phosphatidylinositol 4-phosphate (PtdIns4P) in the *trans*-Golgi regions [10] while CERT is

associated with the ER via binding of its diphenylalanine in an acidic tract (FFAT)-motif to VAP [27]. Replacement of a glycine at position 67 to glutamic acid (G67E) in CERT disrupts the PtdIns4P-binding activity of the PH domain [10]. The punctate distribution of the CERT S135P mutant was abrogated by an additional mutation of either G67E (i.e., G67E/S135P) or deletion of the FFAT-motif (i.e., S135P/ΔFFAT) (Fig 5A and 5B). When the intracellular activity of CERT was assessed by both the lysenin sensitivity assay and *de novo* SM synthesis, the enhanced activity of CERT conferred by the S135P mutation was partially blocked by the additional mutation of G67E or ΔFFAT (Fig 5C and 5D). Together with our previous studies showing the analogous behavior of other activated CERT mutants (i.e., S132A and S315E) [12, 28] that exhibited punctate distributions, these results suggested that the constitutively activated CERT mutants commonly display a subcellular punctate distribution, depending on their ability to associate with both Golgi-embedded PtdIns4P and ER-embedded VAP-A. Although the entity of the CERT-associated puncta remains unclear, the common intracellular punctate distribution pattern of various activated CERT mutants may be applicable as a molecular diagnostic assay to assess whether CERT is abnormally activated by *CERT1* mutations.

## Discussion

The novel *CERT1* missense mutation in our patient is located in the SRM, which is also the site of other *CERT1* mutations previously found in patients with ID (Fig 2D) [4, 5, 7]. Biochemical characterization revealed that amino acid substitutions at S135 in CERT resulted in the loss of SRM hyperphosphorylation, thereby abnormally activating the CERT mutants. Thus, ID patients with activated CERT might be treated with pharmacological inhibitors of CERT activity. To this end, small chemicals that inhibit the inter-membrane ceramide transfer activity of CERT in cultured cells have been developed [29–31].

Our patient exhibited multiple congenital anomalies including cerebral leukodystrophy, skeletal abnormalities and distinctive facial features (Fig 1). It is currently infeasible to determine whether these traits are typical in patients with abnormally activated CERT because clinical features of patients with *CERT1* mutations have not been reported in detail [4–7, 9, 24, 25]. One case of a patient with a substitution of a glycine at position 243 for an arginine (G243R) in CERT exhibited unilateral renal aplasia as well as abnormal brain structures, which included dilated ventricles and periventricular leukomalacia [6]. Additional clinical case reports are required to elucidate whether congenital anomalies other than ID are typical features of *CERT1* mutation. Considering that CERT mutations have been identified as the cause of ASD outside of SRM [24, 25], future case accumulation may indicate that CERT is the causative gene of neurodevelopmental disorders, including ID, developmental delay, and ASD.

Importantly, the ID-inducing missense mutations in *CERT1* described previously have been classified as heterozygous dominant mutations. However, we failed to detect a significant difference in SM synthesis among the trio-derived LCLs by metabolic lipid-labeling experiments and lysenin tolerance assay (Fig 3C and 3D). Although we currently lack a conclusive answer to this discrepancy, there are a few possible explanations: (1) both the metabolic lipid-labeling experiment and lysenin tolerance assay might not be sensitive enough to detect an enhancement in intracellular CERT activity and (2) LCLs might not be a good cell model to monitor changes in the activity of CERT with ID-inducing mutations.

Our patient exhibited delayed cerebrum myelination. Galactosylceramide (GalCer) and its sulfonated metabolite are abundant in the brain and testis, and are essential for the electrophysiological function and stability of the myelin sheath [32, 33]. Since ceramide is a common precursor for SM, glucosylceramide, and GalCer, the enhanced production of SM

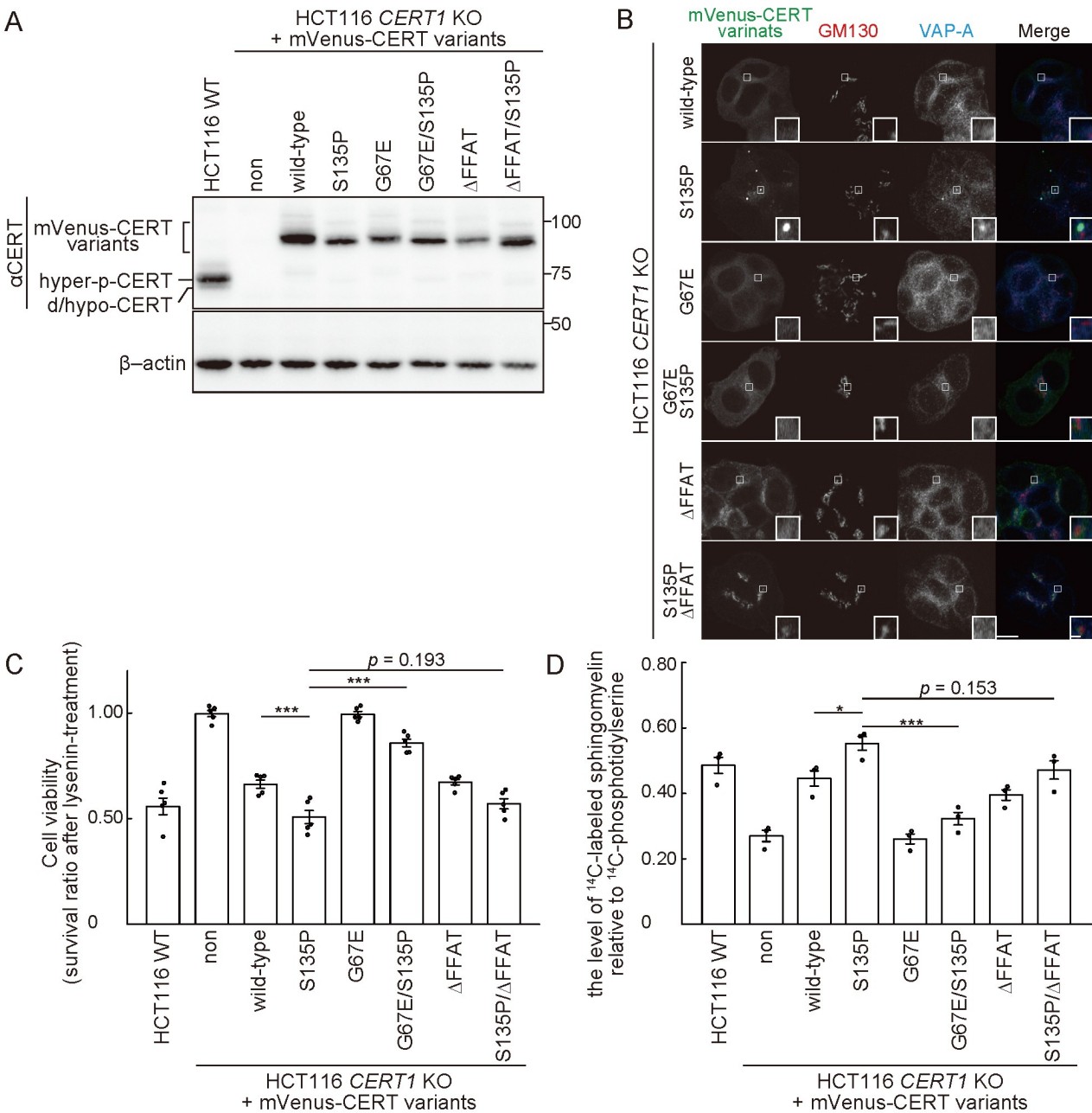

**Fig 5. The S135 mutations in CERT enhance its *de novo* SM synthesis activity.** (A) WT and *CERT1* KO HCT116 cells stably expressing various mVenus-CERT constructs were analyzed by Western blotting with the indicated primary antibodies. (B) *CERT1* KO HCT116 cells stably expressing various mVenus-CERT constructs were fixed and analyzed by immunofluorescence microscopy. GM130 and VAP-A were immuno-stained with specific antibodies. The data are representative of two independent experiments. The *areas enclosed by rectangles* are enlarged (insets). Scale bars, 10 μm and 1 μm (inset). (C) WT and *CERT1* KO HCT116 cells stably expressing various mVenus-CERT constructs were cultured with 250 ng/ml of lysenin for 1 hr. Next, cell viability was measured with an LDH cytotoxicity assay. The data comprise the mean ± SEM; n = 3 (***, $p < 0.001$). (D) The lipids in WT and *CERT1* KO HCT116 cells stably expressing various mVenus-CERT constructs were metabolically labelled with L-[U-$^{14}$C]serine. The labeled lipids were analyzed by TLC and visualized using an image analyzer. The data comprise the mean ± SEM; n = 3 (*, $p < 0.05$; ***, $p < 0.001$).

due to constitutively activated CERT might induce a shortage of the precursor ceramide for GalCer, thereby inhibiting myelination. In order to test this hypothesis, further studies including the generation of a model ID animal with a heterozygous *CERT1* mutation is required.

In conclusion, the present study provided a possible molecular basis for not only new diagnostics but also a conceivable pharmaceutical intervention for ID disorders caused by gain-of-function mutations in *CERT1*.

## Supporting information

**S1 Fig. Three major isoforms of *CERT1* transcripts and their protein products.** (A) The structures of *CERT1* transcripts (isoforms 1–3) whose exons are indicated by gray boxes (top panel) and their protein products (CERT/L, CERT, and CERT/VL) (bottom panel). CERT has a PH domain, a SRM, a FFAT motif, and a START domain. Isoform 1 (NM_0031361.3) contains 17 exons (2–11 and 13–19) and is translated into CERT (NP_112729.1) (top panel). Isoform 2 (NM_005713.3) contains 18 exons (2–19) and is translated into CERT/L (NP_005704.1) (middle panel). Isoform 3 (NM_001130105) contains 19 exons (1–19) and is translated into CERT/VL (NP_001123577) (bottom panel).
(PDF)

**S2 Fig. RNA-sequencing data of *CERT1* transcripts from human tissues using the GTEx database.** (A) Exon-level expression data of *CERT1* (https://gtexportal.org/home/transcriptPage). The heatmap summarizes the median read counts per base of each exon across all tissues. This indicated that the transcript containing exon1 is barely detected in all tissues except for testis and Epstein-Barr virus-transformed lymphocytes. In most tissues, RNA-seq short reads from exon12 were detected at lower frequencies than those from exon11 or exon13. (B) Junction expression data of *CERT1* (https://gtexportal.org/home/transcriptPage). The heatmap summarizes the median raw read counts of junctions from individual isoforms. This demonstrated that read counts of junction21 connecting exon1 with exon2 were barely detected, which indicated that isoform 3 of *CERT1* is not expressed in most human tissues. The read counts of junction10 connecting exon11 with exon13 were detected at higher frequencies than junction9 or junction11, which indicated that the isoform 1 of *CERT1* is the predominant transcript in most human tissues.
(PDF)

**S3 Fig. Establishment of *CERT1* KO HCT116 cells.** (A) Diagram of the location of gRNA-targeted sequences of human *CERT1*. The 5'-untranslated regions (UTRs) and coding regions are shown as white and black rectangles, respectively. The red lines represent the first and second methionines. In the WT allele, the protospacer adjacent motif (PAM) sequence is shown in bold. The red highlighted sequence (ATG) indicates the second methionine within exon2. The gRNA-targeted sequence is underlined. A 1 bp insertion that generates a frame-shift mutation is shown in blue in both *CERT1* alleles from the *CERT1* KO cell line. (B) Cell lysates prepared from WT and *CERT1* KO HCT116 cells were incubated with or without λPPase and analyzed by Western blotting with the indicated primary antibodies. (C) WT and *CERT1* KO HCT116 cells stably expressing various mVenus-CERT constructs were lysed and subjected to Western blotting analysis with the indicated primary antibodies. (D) WT and *CERT1* KO HCT116 cells stably expressing the indicated mVenus-CERT variants were cultured with L-[U-$^{14}$C]serine for 16 hr. Metabolically labelled lipids were separated by TLC analysis and visualized using an image analyzer. A representative image is shown. PC, putative phosphatidylcholine. Note that serine is directly incorporated into PS by the base exchange reaction with PC, and labelled PS is converted to PE by decarboxylation [34, 35]. Metabolic labeling of PC with L-[U-$^{14}$C]serine can occur by two different pathways [34, 35]. One pathway occurs via *N*-methylation of labelled PE, and the other pathway occurs via the conversion of serine to pyruvate, which is then anabolized to fatty acids via acetyl-CoA, followed by the synthesis of PC

from the labeled fatty acids.
(PDF)

**S4 Fig. CERT/L rescues the *de novo* SM synthesis defect in *CERT1* KO HCT116 cells.** (A) WT and *CERT1* KO HCT116 cells stably expressing either mVenus-CERT or mVenus-CERT/L were lysed and analyzed by Western blotting with the indicated primary antibodies. (B) The lipids from WT and *CERT1* KO HCT116 cells stably expressing either mVenus-CERT or mVenus-CERT/L were metabolically labelled with L-[U-$^{14}$C]serine. The labeled lipids were analyzed by TLC and visualized using an image analyzer. The data comprise the mean ± SEM; n = 3 ($^{***}$, $p < 0.001$).
(PDF)

**S5 Fig. Original uncropped and unadjusted images underlying all blot.**
(PDF)

**S1 Table. Clinical information of the patient in the present study.**
(XLSX)

## Acknowledgments

We would like to thank Dr. Ituro Inoue (Division of Human Genetics, National Institute of Genetics, Mishima, Japan) for WES analysis of the patient. We also thank Drs. Kumagai Keigo, Shota Sakai, and Toshiyuki Yamaji (Department of Biochemistry and Cell Biology, National Institute of Infectious Diseases, Tokyo) for their helpful comments on the biochemical and cell biological parts of this work.

## Author Contributions

**Conceptualization:** Hiroaki Murakami, Kentaro Hanada.

**Data curation:** Hiroaki Murakami, Norito Tamura, Yumi Enomoto, Kenji Kurosawa, Kentaro Hanada.

**Formal analysis:** Hiroaki Murakami, Norito Tamura, Yumi Enomoto, Kentaro Shimasaki.

**Funding acquisition:** Norito Tamura, Kenji Kurosawa, Kentaro Hanada.

**Investigation:** Hiroaki Murakami, Norito Tamura, Kentaro Shimasaki.

**Methodology:** Norito Tamura, Kentaro Shimasaki.

**Project administration:** Kenji Kurosawa, Kentaro Hanada.

**Resources:** Kenji Kurosawa, Kentaro Hanada.

**Software:** Hiroaki Murakami, Yumi Enomoto.

**Supervision:** Kenji Kurosawa, Kentaro Hanada.

**Validation:** Hiroaki Murakami, Kentaro Hanada.

**Writing – original draft:** Hiroaki Murakami, Norito Tamura, Kentaro Hanada.

**Writing – review & editing:** Norito Tamura, Kenji Kurosawa, Kentaro Hanada.

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
