## [Decision Letter · Decision Letter 0]

16 Sep 2020

PONE-D-20-26020

Intellectual disability-associated gain-of-function mutations in CERT1 that encodes the ceramide transport protein CERT

PLOS ONE

Dear Dr. Hanada,

Thank you for submitting your manuscript to PLOS ONE. After careful consideration, we feel that it has merit but does not fully meet PLOS ONE’s publication criteria as it currently stands. Therefore, we invite you to submit a revised version of the manuscript that addresses the points raised during the review process.

We look forward to receiving your revised manuscript.

Kind regards,

Ajay Srivastava, PhD

Academic Editor

PLOS ONE

Journal Requirements:

2.PLOS ONE now requires that authors provide the original uncropped and unadjusted images underlying all blot or gel results reported in a submission’s figures or Supporting Information files. This policy and the journal’s other requirements for blot/gel reporting and figure preparation are described in detail at https://journals.plos.org/plosone/s/figures#loc-blot-and-gel-reporting-requirements and https://journals.plos.org/plosone/s/figures#loc-preparing-figures-from-image-files. When you submit your revised manuscript, please ensure that your figures adhere fully to these guidelines and provide the original underlying images for all blot or gel data reported in your submission. See the following link for instructions on providing the original image data: https://journals.plos.org/plosone/s/figures#loc-original-images-for-blots-and-gels.

3.PLOS requires an ORCID iD for the corresponding author in Editorial Manager on papers submitted after December 6th, 2016. Please ensure that you have an ORCID iD and that it is validated in Editorial Manager. To do this, go to ‘Update my Information’ (in the upper left-hand corner of the main menu), and click on the Fetch/Validate link next to the ORCID field. This will take you to the ORCID site and allow you to create a new iD or authenticate a pre-existing iD in Editorial Manager. Please see the following video for instructions on linking an ORCID iD to your Editorial Manager account: https://www.youtube.com/watch?v=_xcclfuvtxQ

4.We note that Figure [1] includes an image of a patient / participant  in the study. 

6. Please include a separate caption for each figure in your manuscript.

<h1>** **</h1>

Additional Editor Comments (if provided):

This manuscript has now undergone peer review and as you will see both of the reviewers have expressed enthusiasm about this manuscript. I would request that you incorporate the suggestions made by the reviewers and then resubmit. Thanks.

Reviewers' comments:

Reviewer's Responses to Questions

**Comments to the Author**

1. Is the manuscript technically sound, and do the data support the conclusions?

Reviewer #1: Yes

Reviewer #2: Yes

2. Has the statistical analysis been performed appropriately and rigorously? 

Reviewer #1: Yes

Reviewer #2: Yes

3. Have the authors made all data underlying the findings in their manuscript fully available?

Reviewer #1: Yes

Reviewer #2: Yes

4. Is the manuscript presented in an intelligible fashion and written in standard English?

Reviewer #1: Yes

Reviewer #2: Yes

5. Review Comments to the Author

Reviewer #1: This study characterizes several de novo mutations in CERT that are linked to a broad spectrum neurodevelopmental disorder in children. Here (and conjunction with previous reports) the authors characterize the effect of these SRM point mutations on CERT activity in patient and KO cells. The experiments indicate that S315 mutations cause constitutive dephosphorylation and activation of CERT, consistent with localization to cellular puncta that appear to be contact sites between he Golgi and ER. The effect of mutant expression is to increase SM synthesis in a CERT-null cell but did not affect SM synthesis in immortalized B cells from patient and patents. Overall an interesting and convincing characterization of these novel mutations.

I have no major concerns with the study with respect to design and data interpretation. There are a few typos and awkward sentences here and there but generally clearly written (Line 33 in abstract –basis not bases).

The one point that should be addressed is the author’s focus on the mutations as a cause intellectual disability (ID) when it is clear that the S315P and other S315 mutations are linked to a broad spectrum neurodevelopmental disorder. ID is only one component of this disorder, which the authors clearly slate on line 402 and 403. The title and some passages in the text give the impression that ID is the only or major defect associated with the mutation when this is not the case. I suggest that the title, as well as some instances in the abstract and introduction, be changed to refer to the spectrum of neurological and other defects associated with CERT SRM mutations. It is evident that the CERT mutations cause demyelination and likely other brain malformations. Thus the defect associated with CERT mutations should be referred to as a “neurodevelopmental disorder” (or something similar) that captures the range of symptoms, including but not limited to intellectual disability.

Reviewer #2: Previous work identified various de novo mutations in CERT1 that are strongly associated with the induction of intellectual disability (ID). In the present study, Murakami report the identification of a novel CERT variant with a S135P substitution in a patient with severe ID. Biochemical experiments with patient-derived cell lines and cells that express pathogenic CERT variants in the absence of the wild-type protein indicate that S135P and several previously identified ID-associated mutants give rise to a constitutively activated CERT protein that is unable to undergo hyperphosphorylation and renders cells more sensitive to lysenin, a sphingomyelin-binding toxin.

Overall, this is a compelling, interesting and well conducted study that provides fresh insight into the molecular basis of ID linked to mutations in CERT. The data are of high quality and the manuscript is well written. I only have a few minor comments.

1) The version of the MS that I reviewed lacked figure legends. To avoid delays, I will be happy to have a closer look at those in a next round of review.

2) All ID-inducing missense mutations in CERT1 described so far, including the one reported in this study, have been classified as heterozygous dominant mutations. However, the authors were unable to detect any significant differences in SM biosynthesis between the patient- and parent-derived immortalized B-cell lines by metabolic lipid labeling. On p. 27 (lines 415-419) they anticipate that metabolic lipid labeling might not be sensitive enough to detect an enhanced CERT activity or that the immortalized B-cell lines are not a good cell model to monitor changes in CERT activity. I wonder whether they checked whether these cell lines showed any differences in the lysenin sensitivity assay, as this assay seems more sensitive than metabolic lipid labeling (Fig. 4B and C).

3) On p. 21 the authors state: “Since the CERT phosphorylation status was only partially shifted to the de/hypophosphorylated form of the ID patient, this indicated that the patient’s mutation is heterozygous….”. As the heterozygous nature of the mutation can only be directly inferred from the sequencing data and not from the WB analysis, I would rephrase this sentence to “These results are in line with the finding that the patient’s mutation is heterozygoes and suggest that the CERT S135P mutant is incapable of becoming hyper-phosphorylated”.

4) The authors should rephrase the last sentence of the Abstract. More suitable would be if they write: “These findings provide a possible molecular basis for not only new diagnostics but also a conceivable pharmaceutical intervention for ID disorders cause by gain-of-function mutations in CERT1.”

5) p. 20, line 303: “…using trio-derived B-cell lines that was established…” should read “…using trio-derived B-cell lines that were established…”

6. PLOS authors have the option to publish the peer review history of their article (what does this mean?). If published, this will include your full peer review and any attached files.

Reviewer #1: **Yes: **Neale Ridgway

Reviewer #2: No

---

## [Author Response · Author response to Decision Letter 0]

26 Nov 2020

Responses to Reviewers’ comments

Major changes are highlighted in red (in the file of Revised Manuscript with Track Changes).

Dr. Kentaro Shimasaki performed new experiments for the revision of the manuscript. All authors have agreed that he become a co-author in the revised manuscript. (p.1 Line 6)

Reviewer #1

General Comment: This study characterizes several de novo mutations in CERT that are linked to a broad spectrum neurodevelopmental disorder in children. Here (and conjunction with previous reports) the authors characterize the effect of these SRM point mutations on CERT activity in patient and KO cells. The experiments indicate that S315 mutations cause constitutive dephosphorylation and activation of CERT, consistent with localization to cellular puncta that appear to be contact sites between he Golgi and ER. The effect of mutant expression is to increase SM synthesis in a CERT-null cell but did not affect SM synthesis in immortalized B cells from patient and patents. Overall an interesting and convincing characterization of these novel mutations.

Response: Thank you very much for your favorable evaluation to our manuscript, and also for helpful specific comments depicted below.

Comment: I have no major concerns with the study with respect to design and data interpretation. There are a few typos and awkward sentences here and there but generally clearly written (Line 33 in abstract –basis not bases).

The one point that should be addressed is the author’s focus on the mutations as a cause intellectual disability (ID) when it is clear that the S315P and other S315 mutations are linked to a broad spectrum neurodevelopmental disorder. ID is only one component of this disorder, which the authors clearly slate on line 402 and 403. The title and some passages in the text give the impression that ID is the only or major defect associated with the mutation when this is not the case. I suggest that the title, as well as some instances in the abstract and introduction, be changed to refer to the spectrum of neurological and other defects associated with CERT SRM mutations. It is evident that the CERT mutations cause demyelination and likely other brain malformations. Thus the defect associated with CERT mutations should be referred to as a “neurodevelopmental disorder” (or something similar) that captures the range of symptoms, including but not limited to intellectual disability.

Response: According to the comments by you and another reviewer, we rephrased the last sentence of abstract as followed: These findings provide a possible molecular basis for not only new diagnostics but also a conceivable pharmaceutical intervention for ID disorders caused by gain-of-function mutations in CERT1.

Thank you for your important comment that ID is the only or major defect associated with the mutation when this is not the case. The reviewer’s view is right on our patient with the S135P variant. However, for a previously reported patient with S135C variant, clinical or neurological features other than ID are unclear (not described in literature. Therefore, we can’t refer to the common spectrum of neurological and other defects associated with CERT SRM mutations at the present. To present the view pointed by the reviewer, we discussed as followed: “Considering that CERT mutations have been identified as the cause of ASD outside of SRM [24, 25], future case accumulation may indicate that CERT is the causative gene of neurodevelopmental disorders, including ID, developmental delay, and ASD.” (p. 30, Line 472-475)

Other revisions:

According to the editorial rule of PLoS One, we stated “Written informed consent (as outlined in PLOS consent form) was obtained from the patient’s parents, which included consent for the pictures appearing in the manuscript.” (p. 5, Line 74-76).

We improved the description of the procedures for “lysenin tolerance assay” as followed: In our assay, the cell viability (“survival ratio after lysenin-treatment”) was defined as the following formula: 1-(A1-A2)/(A3-A2). Here, A1, A2, and A3 represent LDH activity (measured by absorbance) from lysenin-treated cells, un-treated control cells, and detergent-treated cells, respectively (p. 9, Line 155-158). The caption for the y-axis of the panels for lysenin-sensitivity (Fig. 4C and Fig. 5C) was also changed from "Cell viability (relative to lysenin-treated cells)” to “Cell viability (survival ratio after lysenin-treatment)”. 

Reviewer #2: 

General comment: Previous work identified various de novo mutations in CERT1 that are strongly associated with the induction of intellectual disability (ID). In the present study, Murakami report the identification of a novel CERT variant with a S135P substitution in a patient with severe ID. Biochemical experiments with patient-derived cell lines and cells that express pathogenic CERT variants in the absence of the wild-type protein indicate that S135P and several previously identified ID-associated mutants give rise to a constitutively activated CERT protein that is unable to undergo hyperphosphorylation and renders cells more sensitive to lysenin, a sphingomyelin-binding toxin.

Overall, this is a compelling, interesting and well conducted study that provides fresh insight into the molecular basis of ID linked to mutations in CERT. The data are of high quality and the manuscript is well written. I only have a few minor comments.

Response: Thank you very much for your favorable evaluation to our manuscript, and also for constructive specific comments. We attempted to answer all specific comments as described below.

Comment 1) The version of the MS that I reviewed lacked figure legends. To avoid delays, I will be happy to have a closer look at those in a next round of review.

Response: We are very sorry for the big mistake. The legends to figures of the main text were added in the Result section, not after the Reference section, according to the PLoS One style (each Figure legend was highlighted in red, respectively). We deeply appreciate so high scientific capacity of both reviewers who were able to evaluate the original manuscript even without figure legends.

Comment 2) All ID-inducing missense mutations in CERT1 described so far, including the one reported in this study, have been classified as heterozygous dominant mutations. However, the authors were unable to detect any significant differences in SM biosynthesis between the patient- and parent-derived immortalized B-cell lines by metabolic lipid labeling. On p. 27 (lines 415-419) they anticipate that metabolic lipid labeling might not be sensitive enough to detect an enhanced CERT activity or that the immortalized B-cell lines are not a good cell model to monitor changes in CERT activity. I wonder whether they checked whether these cell lines showed any differences in the lysenin sensitivity assay, as this assay seems more sensitive than metabolic lipid labeling (Fig. 4B and C).

Response: Thank you for your helpful advice. According to the advice, we have examined the sensitivity of the trio LCLs to lysenin, an SM-binding cytolysin. The patient-derived LCLs did not exhibit a clear difference in the lysenin-sensitivity when compared with the parents’ controls (the actual data are shown below for reviewers only). Although the father-derived LCLs may have a trend of more tolerance, we noticed that the growth of father-derived LCLs is slower than that of patient- and mother-derived LCLs under normal culture conditions. This might affect the lysenin-sensitivity of the father-derived LCLs. Therefore, we consider that it is inappropriate to include the new experimental data in the revised manuscript at the current stage.

Trio LCLs were cultured with 250 ng/ml of lysenin for 1 hr. Next, cell viability was measured with a LDH cytotoxicity assay kit. The data comprise the mean ± SEM; n=3 (n.s., not significant).

The reviewer’s comment also let us notice that the description of the procedures for “lysenin tolerance assay” was confusing. To improve the description, we rewrote it using a formula as followed: In our assay, the cell viability (“survival ratio after lysenin-treatment”) was defined as the following formula: 1-(A1-A2)/(A3-A2). Here, A1, A2, and A3 represent LDH activity (measured by absorbance) from lysenin-treated cells, un-treated control cells, and detergent-treated cells, respectively (p. 9, L 155-158). The caption for the y-axis of the panels for lysenin-sensitivity (Fig. 4C and Fig. 5C) was also changed from "Cell viability (relative to lysenin-treated cells)” to “Cell viability (survival ratio after lysenin-treatment)”.

Comment 3) On p. 21 the authors state: “Since the CERT phosphorylation status was only partially shifted to the de/hypophosphorylated form of the ID patient, this indicated that the patient’s mutation is heterozygous….”. As the heterozygous nature of the mutation can only be directly inferred from the sequencing data and not from the WB analysis, I would rephrase this sentence to “These results are in line with the finding that the patient’s mutation is heterozygoes and suggest that the CERT S135P mutant is incapable of becoming hyper-phosphorylated”.

Response: We agree with the reviewer’s consideration and rephrased the sentence as the reviewer commented. (p. 22, Line 334-336)

Comment 4) The authors should rephrase the last sentence of the Abstract. More suitable would be if they write: “These findings provide a possible molecular basis for not only new diagnostics but also a conceivable pharmaceutical intervention for ID disorders caused by gain-of-function mutations in CERT1.”

Response: According to the comment by the reviewer, the last sentence of abstract was revised as followed: These findings provide a possible molecular basis for not only new diagnostics but also a conceivable pharmaceutical intervention for ID disorders caused by gain-of-function mutations in CERT1. (p. 2, Line 33-35). In addition, the rephrased sentence was also added as the last sentence of the discussion section: “In conclusion, the present study provided a possible molecular basis for not only new diagnostics but also a conceivable pharmaceutical intervention for ID disorders caused by gain-of-function mutations in CERT1.” (p. 31, Line 493-495).

Comment 5) p. 20, line 303: “…using trio-derived B-cell lines that was established…” should read “…using trio-derived B-cell lines that were established…”

Response: Corrected as commented.

Other revisions:

According to the editorial rule of PLoS One, we stated “Written informed consent (as outlined in PLOS consent form) was obtained from the patient’s parents, which included consent for the pictures appearing in the manuscript.” (p. 5, Line 74-76).

---

## [Decision Letter · Decision Letter 1]

2 Dec 2020

Intellectual disability-associated gain-of-function mutations in CERT1 that encodes the ceramide transport protein CERT

PONE-D-20-26020R1

Dear Dr. Hanada,

We’re pleased to inform you that your manuscript has been judged scientifically suitable for publication and will be formally accepted for publication once it meets all outstanding technical requirements.

Kind regards,

Ajay Srivastava, PhD

Academic Editor

PLOS ONE

Additional Editor Comments (optional):

Reviewers' comments:

Reviewer's Responses to Questions

**Comments to the Author**

1. If the authors have adequately addressed your comments raised in a previous round of review and you feel that this manuscript is now acceptable for publication, you may indicate that here to bypass the “Comments to the Author” section, enter your conflict of interest statement in the “Confidential to Editor” section, and submit your "Accept" recommendation.

Reviewer #1: All comments have been addressed

Reviewer #2: All comments have been addressed

2. Is the manuscript technically sound, and do the data support the conclusions?

Reviewer #1: Yes

Reviewer #2: Yes

3. Has the statistical analysis been performed appropriately and rigorously? 

Reviewer #1: Yes

Reviewer #2: Yes

4. Have the authors made all data underlying the findings in their manuscript fully available?

Reviewer #1: Yes

Reviewer #2: Yes

5. Is the manuscript presented in an intelligible fashion and written in standard English?

Reviewer #1: Yes

Reviewer #2: Yes

6. Review Comments to the Author

Reviewer #1: (No Response)

Reviewer #2: The authors addressed all my concerns thoughtfully and to the best of their ability. XXXXXXXXXXXXXXXXXXXXXX

7. PLOS authors have the option to publish the peer review history of their article (what does this mean?). If published, this will include your full peer review and any attached files.

Reviewer #1: No

Reviewer #2: **Yes: **Joost Holthuis

---

## [Editor Report · Acceptance letter]

10 Dec 2020

PONE-D-20-26020R1 

Intellectual disability-associated gain-of-function mutations in *CERT1* that encodes the ceramide transport protein CERT 

Dear Dr. Hanada:

I'm pleased to inform you that your manuscript has been deemed suitable for publication in PLOS ONE. Congratulations! Your manuscript is now with our production department. 

Kind regards, 

on behalf of

Dr. Ajay Srivastava 

Academic Editor

PLOS ONE